# Maternal Milk Provision in the Neonatal Intensive Care Unit and Mother–Infant Emotional Connection for Preterm Infants

**DOI:** 10.3390/children9020296

**Published:** 2022-02-21

**Authors:** Clare Viglione, Sara Cherkerzian, Wendy Timpson, Cindy H. Liu, Lianne J. Woodward, Mandy B. Belfort

**Affiliations:** 1Department of Pediatric Newborn Medicine, Brigham and Women’s Hospital, Boston, MA 02115, USA; scherkerzian@bwh.harvard.edu (S.C.); chliu@bwh.harvard.edu (C.H.L.); lianne.woodward@canterbury.ac.nz (L.J.W.); mbelfort@bwh.harvard.edu (M.B.B.); 2Department of Pediatrics, Harvard Medical School, Boston, MA 02115, USA; 3Department of Neonatology, Beth Israel Deaconess Medical Center, Harvard Medical School, Boston, MA 02131, USA; wtimpson@bidmc.harvard.edu

**Keywords:** preterm, breastfeeding, maternal milk, mother–infant connection, Neonatal Intensive Care Unit (NICU)

## Abstract

Maternal milk (MM) intake during neonatal intensive care unit (NICU) hospitalization is associated with improved neurodevelopment in preterm infants. Underlying mechanisms may include stronger mother–infant emotional connection. This paper examines associations between MM provision in the NICU with maternal connection to her infant using three factors validated in our sample: maternal sensitivity, emotional concern, and positive interaction/engagement. We studied 70 mothers of infants born <1500 g and/or <32 weeks’ gestation. Associations between MM provision and mother–infant connection were modeled using median regression adjusted for clustering. Mothers who provided exclusive MM (i.e., 100% MM, no other milk) reported higher levels of maternal sensitivity by a median score of 2 units (β = 2.00, 95% CI: 0.76, 3.24, *p* = 0.002) than the mixed group (i.e., MM < 100% days, other milk ≥1 days), as well as greater emotional concern (β = 3.00, 95% CI: −0.002, 6.00, *p* = 0.05). Among mothers of very preterm infants, greater milk provision was associated with greater maternal sensitivity, but also with greater emotional concern about meeting the infant’s needs. These findings highlight the importance of supporting MM provision and early infant care as an integrated part of lactation support. The findings may also provide insight into links between MM provision in the NICU and infant neurodevelopment.

## 1. Introduction

There is clear evidence to support the importance of maternal breastfeeding for child cognitive and socioemotional development, but the specific mechanisms underlying this association are less well understood [1]. Since infants born very preterm (<32 weeks gestational age) are at a heightened developmental risk, identifying early modifiable factors that might help mitigate this risk is crucial [2,3]. Consistent with findings in the full-term population, preterm infants fed a predominantly human milk diet demonstrate improved cognition and behavior later in life [4,5,6]. Although nutritional and non-nutrient bioactive components of maternal milk (MM) likely play an important role in improving brain development for the preterm-born infant, an alternative or complementary explanation is that provision of maternal milk may also offer increased opportunities for mother–infant connection and attachment, which in turn contribute to positive neurodevelopment, especially socioemotional development [7].

Mother–infant connection is related to attachment, described by Bowlby as a relationship whereby children are strongly disposed to seek proximity and contact with their primary caregiver, and who do so in stressful situations [8,9]. In full-term infant populations, infant breastfeeding is associated with more secure mother–child attachment and greater maternal responsiveness [10]. Breastfeeding mothers of term infants also spend more time with their offspring outside of the feeding episode compared with bottle feeding mothers and are more likely to engage in mutual touch [11]. Moreover, maternal behaviors such as tactile stimulation, mother’s gaze, and mutual touch are significantly more frequent during a session of breastfeeding than a session of bottle feeding [12]. Lastly, breastfeeding has been linked to a heightened response to infant cues in maternal brain regions implicated in bonding and empathy [13]. This response may be mediated by the release of oxytocin during breastfeeding, as oxytocin-induced neural activations facilitate greater maternal sensitivity [14].

Overall, available data lend support to the hypothesis that breastfeeding may positively influence mother–infant connection, yet it remains unknown whether maternal milk provision, without direct breastfeeding, also predicts mother–infant connection [10,11,12,13,14]. This gap is of particular relevance to mothers of infants born very preterm, whose infants are unable to feed at the breast for the first several months of life due to immaturity of their oral–motor feeding skills and preterm birth-associated comorbidities. In this study, we aimed to (1) examine the underlying factor structure of a series of scales designed to assess the quality of early mother–infant relations in the neonatal intensive care unit (NICU), and (2) assess associations between maternal milk provision during neonatal hospitalization and maternal reports of the quality of their connection with their newborn infant, taking into consideration clustering among study sites as well as potential confounding by gestational age and maternal education.

## 2. Materials and Methods

### 2.1. Study Sample

Data were drawn from the Parenting in the NICU study, a cross-sectional survey conducted at two academic Level III NICUs in Boston (Brigham and Women’s Hospital (BWH) and Beth Israel Deaconess Medical Center (BIDMC)) between January 2015 and December 2017. Ethics approval was obtained from the Partners Healthcare IRB on 27 January 2015. Parents provided written, informed consent to participate.

The Parenting in the NICU study sample consisted of 300 English-speaking families with infants born at BWH or BIDMC, hospitalized for at least one week, and recruited close to NICU discharge (see Figure 1). At infant discharge, families participated in a 45 min structured interview centered on several domains including physical and mental health, the experience of being a parent in the NICU, and family social circumstances. Inter-interviewer reliability checks were undertaken at regular intervals for 10% to 20% of study participants. The analysis for this paper involved 70 families from either the BIDMC or BWH NICU with singleton infants born <32 weeks’ gestation or <1500 g with complete maternal milk data and who consented to medical record review and discharge interview.

### 2.2. Measures

#### 2.2.1. Maternal Milk Intake

Maternal milk intake data were extracted from the electronic medical record and categorized using methodology developed for a statewide human milk collaborative in which both hospitals participated, as previously described [15]. Briefly, using data regarding the feeding of maternal milk and/or other milk (donor, formula) (yes/no) collected once per week over the entire hospitalization, we categorized MM provision as: (1) maternal milk fed on 100% of days no other milk fed (“exclusive MM”); (2) maternal milk fed on 100% of days, other milk fed on ≥1 day (“predominant MM”); and (3) maternal milk fed on <100% of days, other milk fed on ≥1 days (“mixed”) [15].

#### 2.2.2. Mother–Infant Connection

A major focus of the Parenting in the NICU Study was to understand maternal engagement with the infant’s care during their hospitalization and any parenting issues that arose. Two measures assessing aspects of maternal perceived mother–infant connection were administered. The first measure consisted of a 19 item Maternal Attachment scale by Nagata et al. (2000), which in its original form comprised two subscales (Anxiety and Acceptance). The second measure included 6 items adapted from Melnyk’s Parental Beliefs Scale (PBS), (2014), which comprised three factors: Parental Role Confidence; Parent–Baby Interaction; and Knowledge [16,17,18]. Each item was scored on a Likert scale from 1 (very true) to 5 (very untrue). Items “I worry about my child in many ways when my child is not with me”, “I miss touching or holding my baby when s/he is not with me”, “I am not that interested in my child”, and “I don’t find my baby cute” were reverse coded [16]. The possible range of scores was 25–125, with higher scores indicating greater mother–infant connection.

#### 2.2.3. Psychometrics of Mother–Infant Connection Scales

The reliability and validity of these instruments were first assessed using mother–infant dyads from the Parenting in the NICU Study born <1500 g and/or <32 weeks’ gestation with infant feeding data (*n* = 127). In examining the psychometric properties of the Nagata Maternal Attachment scale in our sample, we noted that the overall Cronbach coefficient alpha was low (raw = 0.41, standardized = 0.22), consistent with more than one latent dimension. Within the Anxiety subscale, the Cronbach coefficient alpha was high (raw = 0.79, standardized = 0.80), but low for the Acceptance subscale (raw = 0.23, standardized = 0.40). Within the 6 items from Melnyk’s PBS, the Cronbach coefficient alpha was low (raw = 0.39, standardized = 0.59); however, excluding the item “Uncomfortable when baby unsettled/demanding” increased the Cronbach coefficient alpha to a level of acceptable internal consistency (raw = 0.71, standardized = 0.76). These findings prompted us to conduct a factor analysis of all items from both scales (please see Appendix A for further technical details). Factor analysis identified four potential factors: (1) maternal sensitivity; (2) emotional concern; (3) positive interaction/engagement; and (4) parenting detachment (shown in Figure 2). The study team collectively reviewed the items within each factor for conceptual clarity and consistency, and together developed factor labels. The items loading onto the first three factors (maternal sensitivity, emotional concern, and positive interaction/engagement) were conceptually robust, but parenting detachment (i.e., factor 4) appeared conceptually weak. The negative loading of the item, “I am not that interested in my child” was confusing, and the factor only accounted for 1.7% of the variance. Accordingly, the study team excluded factor 4 and retained factors 1–3 for primary analyses. Lower factor scores suggested greater sensitivity, greater emotional concern, and greater interaction/engagement with infant.

#### 2.2.4. Covariates

Maternal covariates included hospital location (BIDMC or BWH), age (years), primiparity (Yes/No), medical insurance (Yes/No), socioeconomic status (SES), maternal leave, race/ethnicity, and education. Maternal race/ethnicity were assessed with categorical response options: American Indian/Native American; Asian, Black/African American, Hispanic/Latina, White/Caucasian and other. Maternal education was evaluated with categorical options: high school diploma/GED; some college; college graduate; and advanced professional degree. Hollingshead Four Factor Index was used to assess maternal SES based on four domains: marital status; retired/employed status; educational attainment; and occupational prestige [18]. Higher scores on the Hollingshead index reflect higher SES [19]. Employment leave was classified as follows: less than 3 months of maternal leave; 3 months of leave; more than 3 months; not returning to work; leave until baby comes home from the NICU; not working prior to birth; and undecided. Infant covariates included sex (male/female), gestational age (number of weeks), preterm birth (<32 weeks’ gestation), NICU length of stay (number of days), and birth weight. Birth weight was measured continuously in grams, categorically (weight < 1500 g), and with the Fenton preterm growth reference to facilitate the calculation of z-scores [20].

### 2.3. Statistical Analysis

The infant clinical and family social background characteristics of the three maternal milk provision groups were examined using chi-square for categorical variables and the Kruskal–Wallis test for continuous variables. Factor analysis was conducted to identify the latent structure among the survey items (*n* = 25). Please see Appendix A for additional description of factor analysis. Correlations between study exposure (MM provision category), outcomes (the three mother–infant connection factors), and sample clinical and social background characteristics were assessed using Spearman rank order correlations due to the non-normal distributions among the variables. Associations between milk category (exclusive MM (1) as reference) and mother–infant connection factors were assessed using median regression with robust covariance estimation to address clustering by study site in models unadjusted and adjusted for potential confounding by gestational age at birth and maternal education. Models were run using the qreg2 wrapper for STATA qreg (StataCorp. 2019. Stata Statistical Software: Release 16. College Station, TX: StataCorp LLC.).

## 3. Results

### 3.1. Study Sample

Characteristics of mothers and infants overall and by MM category are shown in Table 1. Among study mothers, 44% (*n* = 28) identified as White and 28% (*n* = 18) as Black. The majority (60%, *n* = 42) had at least a college degree and 31% (*n* = 22) had an advanced professional degree. In terms of MM provision, over half (54%) of the sample were in the mixed MM category, with only 12.9% from the total sample in the predominant MM group. Overall, maternal race/ethnicity did not differ significantly across the MM categories (*p* = 0.1432 (chi-square)). Among mothers who provided exclusive MM (*n* = 23), a disproportionately greater number were White/Caucasian (70%, *n* = 14) compared with women of other race/ethnicity categories (e.g., Black/African American (20%, *n* = 4)).

### 3.2. Descriptive Statistics of Exposure and Outcome

Factor analysis of the maternal interview items identified three relevant factors for primary analyses: (1) maternal sensitivity; (2) emotional concern for baby; and (3) positive interaction/engagement (Figure 2). Descriptive statistics of the factors identified are listed in Table 2, and ranges among the maternal milk categories are listed in Table 3. The distributions of the factors were skewed, and the range of values (maximum–minimum) varied from 26 (emotional concern, where MM is exclusive MM) to 5 (maternal sensitivity, emotional concern, where MM is mixed). Variation in median values among milk provision categories ranged from 6 (interaction/engagement) to 19 (emotional concern).

### 3.3. Social and Background Characteristics Related to Maternal Milk Provision

Results suggest a diverse sample in terms of its range across demographic variables including SES, age, education, and race/ethnicity. Women who provided exclusive MM were older (mean = 35.0 years, SD = 3.7) compared with mothers in the predominant MM (mean = 34.3 years, SD = 5.3) and mixed categories (mean = 30.5 years, SD = 6.7). The majority of mothers who provided exclusive MM had an advanced degree (*n* = 14, 60.9%), whereas only five mothers (13.2%) had an advanced degree in the mixed group. Further, higher SES levels were observed among the exclusive MM group (mean = 55.3; SD = 9.6) compared with the predominant MM (mean = 41.8; SD = 10.7) and mixed categories (mean = 37.4; SD = 12.4). There were no statistically significant differences across milk categories among infant characteristics.

### 3.4. Relationship between Mother–Infant Connection and Potential Covariates

Correlations between factors of mother–infant connection and covariates were small to moderate (Table 4). Significant (*p* < 0.05) correlations were observed between educational attainment and mother–infant connection, with greater education correlated with less interaction/engagement with infant (Factor 3, lower scores suggest greater interaction/engagement). SES exhibited small to moderate correlation with two factors: emotional concern and interaction/engagement, indicating that higher SES relates to greater feelings of emotional concern around caregiving (r_s_ = −0.31, *p* < 0.05) and less interaction/engagement with the newborn (r_s_ = 0.48, *p* < 0.001). We did not adjust for SES in our median regression models due to its high correlation with maternal education (r_s_ = 0.87, *p* < 0.001).

### 3.5. Associations of Maternal Milk Provision with Mother–Infant Connection

Table 5 shows unadjusted and adjusted associations between MM provision category and measures of perceived maternal connection to her infant. In analyses adjusted for potential confounding by gestational age and maternal education, mothers in the exclusive MM category (reference) reported greater maternal sensitivity than mothers in the mixed MM category (2 points higher score in mixed group, 95% CI 0.76, 3.24) and greater emotional concern than mothers in the predominant and mixed MM categories (3 points higher score, 95% CI 1.14, 4.86 for predominant MM; 3 points higher, 95% CI 0.00, 6.00 for mixed MM).

## 4. Discussion

To our knowledge, this is the first study to examine, in a NICU setting, whether maternal milk provision is associated with a mother’s perceived emotional connection to her preterm newborn. Drawing on evidence from other contexts, we hypothesized that maternal milk provision would lead to stronger mother–infant connection. Using three relevant constructs that we validated in our own sample, our main finding was that greater MM provision was associated with aspects of mother–infant connection including greater maternal sensitivity and greater emotional concern for baby.

### 4.1. Main Associations between Maternal Milk Provision Level and Mother–Infant Connection

We observed that greater MM provision was associated with improved maternal sensitivity, a central component of healthy mother–infant attachment. This finding aligns with evidence that mothers who exclusively breastfeed have oxytocin-induced hypothalamic and pituitary neural changes in response to their infant’s cues that help women be attuned to their infants, interpret cues, and strengthen feelings of connection [21,22,23]. Other research links breastfeeding and maternal sensitivity, which in turn, predicts reduced levels of negative affectivity in infant temperament [24]. We also observed that greater MM provision was associated with greater emotional concern. This likely reflects these mothers’ greater concern or anxiety regarding what they think might be best for their infant in the NICU context, including providing maternal milk [25,26]. Mothers in the NICU have reported that pumping and providing breast milk is an act of caring for their infant and facilitates bonding [27,28]. Further, symptoms of state and trait anxiety are not always associated with reduced breastfeeding in NICU settings, as researchers sometimes predict [29]—and maternal worry can even facilitate—more optimal maternal–infant bonding [30]. Our study, taken together with previous studies, supports the hypothesis that MM provision contributes to greater maternal sensitivity, an important predictor of long-term mother–infant attachment. Although preliminary research suggests an association between emotional concern and MM provision, more research with larger samples is needed to clearly interpret these results.

### 4.2. Constructs of Mother–Infant Connection in a Preterm Sample

Using the 25-item, Parenting in the NICU Study survey, we identified three factors of mother–infant connection (1) maternal sensitivity, (2) emotional concern, and (3) positive interaction/engagement. The factors observed in the current study differ from sub-scales derived by Nagata et al., perhaps because the sample with which we applied the Nagata survey is different from that with which it was validated. Differences between the Nagata et al. factors and those observed in the current analysis suggest that the dimensions underlying attachment may differ between mothers of full-term infants and those of preterm infants and that the ability to recognize and interpret infant cues (maternal sensitivity) which is not included in Nagata’s scale, may be a particularly important construct of mother–infant connection among preterm dyads. We expect that our questionnaire may be useful to other researchers interested in measuring perceived maternal connection to her infant in the NICU.

### 4.3. Maternal Milk, Mother–Infant Connection, and Demographic Factors

In this sample, the exclusive MM group included more White mothers (70% identified as White) than the mixed MM category, which also included a higher proportion of Black mothers (31.4% identified as Black, and 31.4% identified as White in the mixed MM group). These disparities in MM provision between Black and White women is consistent with broader public health trends in breastfeeding rates [31], which show that Black infants have a significantly lower rate of any breastfeeding at 3 months (58%) than White infants (73%) [32]. We also found that White mothers report greater positive interaction/engagement with their infants (Table 4) replicating findings that White mothers appear to be more engaged with their infants in the NICU than Black mothers and also with healthcare in general [33,34]. This raises critical questions about structural barriers associated with systemic racism that pose disproportionate impacts on Black mothers’ ability to access quality care, interact and engage with trusted providers including lactation consultants in the NICU, and also initiate and continue pumping MM. Black women, especially those of lower SES, are less likely to initiate breastfeeding [32] and often need to make difficult trade-offs related to income, time, and travel before devoting finite resources to MM provision [35].

### 4.4. Study Strengths and Limitations

Although the Parenting in the NICU Study survey relied on self-report susceptible to social desirability, MM data were monitored and collected in real-time, during intensive clinical care, potentially boosting the reliability and validity of these data. Unfortunately, frequency and duration of kangaroo mother care or skin-to-skin contact were not available for inclusion in analyses; as such, investigating kangaroo care in future studies of maternal milk and mother–infant connection is important. This study adds to the growing literature examining mother–infant emotional connection among preterm infants, as the concept is less clearly understood than those for full term infants [36,37]. Although the design is observational, we addressed confounding bias by evaluating several variables as potential confounders based on the literature and from the conceptual model and adjusting for confounding bias for those variables correlated with both exposure and outcome. We also adjusted for robust covariance estimation to address clustering by site. Nonetheless, residual confounding by unknown factors may still be possible.

## 5. Conclusions

In this study of MM provision in a diverse sample of mothers of preterm infants, we (1) report a psychometrically sound survey assessing maternal perceptions of connection to their infant, and (2) found that greater provision of MM may provide a mother with improved maternal sensitivity and may also be associated with greater emotional concern for meeting the child’s needs. These findings provide insight into previously established links between MM provision in the NICU and infant outcomes. They also have implications for optimizing both lactation and mental health supports offered to parents during and after the NICU hospitalization.

## Figures and Tables

**Figure 1 children-09-00296-f001:**
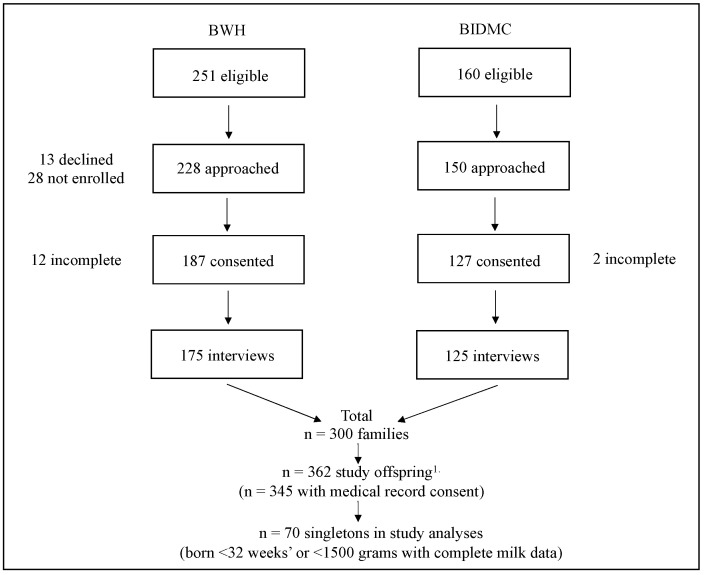
Parenting in the NICU Study Recruitment Flowchart. BWH: Brigham and Women’s Hospital; BIDMC: Beth Israel Deaconess Medical Center. ^1^ There were 370 offspring among 300 families (235 singletons, 120 twins, 15 triplets), 8 twins were not admitted to the NICU for a total of *n* = 362 study infants.

**Figure 2 children-09-00296-f002:**
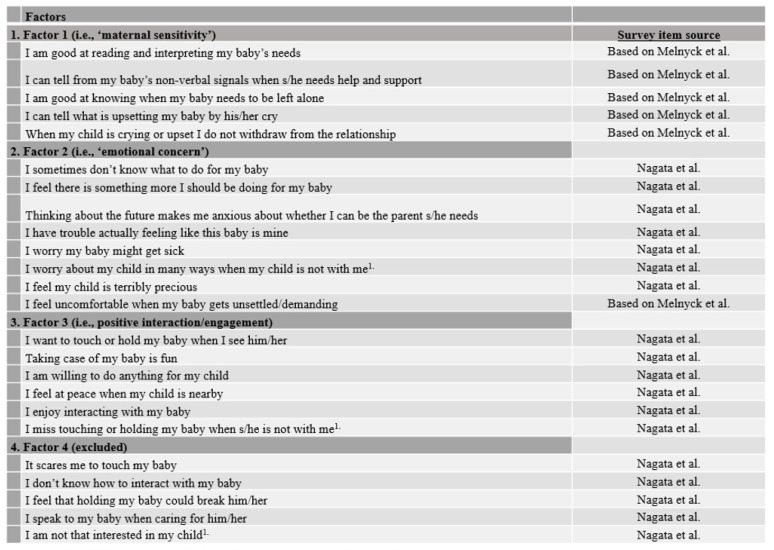
25-item Parenting in the NICU Study Survey [16,17,18]. ^1^ Item “I don’t find my baby cute” from Nagata et al. excluded from factor analysis due to communalities >1; ^1^. Reverse coded.

**Table 1 children-09-00296-t001:** NICU Parenting in the NICU Study (2015–2017). Demographics among mother-infant dyads of singleton infants born <1500 g and/or <32 weeks gestation with infant feeding data, *n* = 70. Results overall and by milk provision category.

Variables	Total (*n* = 70)	Exclusive Maternal Milk (*n* = 23)	Predominant Maternal Milk (*n* = 9)	Mixed Maternal Milk (*n* = 38)	*p* Value
**Maternal Characteristics**					
**Hospital NICU**					0.0293 *
BIDMC	30 (42.9)	5 (21.7)	6 (66.7)	19 (50.0)	
BWH	40 (57.1)	18 (78.3)	3 (33.3)	19 (50.0)	
**Maternal age**					0.0255 *
Nmiss (%)	7 (10.0)	3 (13.0)		4 (10.5)	
Mean ± SD	32.5 ± 6.0	35.0 ± 3.7	34.3 ± 5.3	30.5 ± 6.7	
Min–Max	20.0–46.0	30.0–42.0	25.0–40.0	20.0–46.0	
Median (IQR)	33.0 (30.0–36.0)	33.5 (32.0–37.0)	34.0 (30.0–39.0)	31.0 (26.0–35.0)	
**Maternal race/ethnicity**					0.1432
Missing	6 (8.6)	3 (13.0)		3 (7.9)	
American Indian/Native American	2 (3.1)	1 (5.0)		1 (2.9)	
Asian	2 (3.1)		1 (11.1)	1 (2.9)	
Black/African American	18 (28.1)	4 (20.0)	3 (33.3)	11 (31.4)	
Hispanic/Latina	11 (17.2)		2 (22.2)	9 (25.7)	
White/Caucasian	28 (43.8)	14 (70.0)	3 (33.3)	11 (31.4)	
Other	3 (4.7)	1 (5.0)		2 (5.7)	
**Maternal education level**					0.0006 *
High school or GED	11 (15.7)		1 (11.1)	10 (26.3)	
Some college	17 (24.3)	1 (4.3)	2 (22.2)	14 (36.8)	
College graduate	20 (28.6)	8 (34.8)	3 (33.3)	9 (23.7)	
Advanced degree	22 (31.4)	14 (60.9)	3 (33.3)	5 (13.2)	
**Medical insurance**					
Yes	70 (100)	23 (100)	9 (100)	38 (100)	
**Socioeconomic status (Hollingshead)**					<0.0001 *
Nmiss (%)	2 (2.9)	1 (4.3)		1 (2.6)	
Mean ± SD	43.8 ± 13.9	55.3 ± 9.6	41.8 ± 10.7	37.4 ± 12.4	
Min–Max	17.0–66.0	24.0–66.0	23.5–58.5	17.0–66.0	
Median (IQR)	43.0 (33.5–57.0)	57.8 (54.5–61.0)	40.0 (37.5–43.0)	38.5 (26.0–44.0)	
**Maternal leave**					0.7531
<3 months	4 (6.1)	2 (8.7)		2 (5.7)	
3 months	14 (21.2)	4 (17.4)	3 (37.5)	7 (20.0)	
>3 months	26 (39.4)	13 (56.5)	2 (25.0)	11 (31.4)	
Not returning to work	6 (9.1)	1 (4.3)	1 (12.5)	4 (11.4)	
Not working prior to baby	14 (21.2)	3 (13.0)	2 (25.0)	9 (25.7)	
**Primiparous**					0.2695
No	22 (31.4)	10 (43.5)	3 (33.3)	9 (23.7)	
Yes	48 (68.6)	13 (56.5)	6 (66.7)	29 (76.3)	
**Infant Characteristics**					
**Infant sex**					0.9035
Boy	36 (51.4)	11 (47.8)	5 (55.6)	20 (52.6)	
Girl	34 (48.6)	12 (52.2)	4 (44.4)	18 (47.4)	
**Gestational Age, weeks**					0.1973
Mean ± SD	28.6 ± 2.9	27.8 ± 2.3	29.1 ± 3.6	28.9 ± 3.0	
Min–Max	22.0–39.1	24.0–33.0	22.0–33.6	24.0–39.1	
Median (IQR)	28.1 (27.0–30.3)	28.0 (26.0–29.0)	30.0 (28.1–31.1)	28.3 (27.0–30.9)	
**Gestational age < 32 weeks’ gestation**					0.3187
No	8 (11.4)	1 (4.3)	2 (22.2)	5 (13.2)	
Yes	62 (88.6)	22 (95.7)	7 (77.8)	33 (86.8)	
**Birth weight, g**					0.9785
Mean ± SD	1047.7 ± 317.6	1039.1 ± 352.7	1041.1 ± 296.9	1054.4 ± 308.1	
Min–Max	430.0–1630.0	482.0–1630.0	430.0–1380.0	530.0–1490.0	
Median (IQR)	1050.0 (770.0–1360.0)	970.0 (750.0–1370.0)	1040.0 (910.0–1290.0)	1050.0 (720.0–1375.0)	
**Birth weight, <1500 g**					0.1220
No	2 (2.9)	2 (8.7)			
Yes	68 (97.1)	21 (91.3)	9 (100)	38 (100)	
**Birth weight Z score (Fenton)**					0.2874
Mean ± SD	−0.4 ± 1.1	−0.2 ± 1.2	−0.8 ± 1.1	−0.5 ± 1.1	
Min–Max	−2.3–2.3	−2.3–2.1	−1.8–−1.8	−2.2–2.3	
Median (IQR)	−0.5 (−1.3–0.4)	−0.3 (−1.2–0.9)	−1.2 (−1.4–−0.3)	−0.5 (−1.3–0.2)	
**NICU length of stay, days**					0.7633
Mean ± SD	86.3 ± 35.3	89.7 ± 28.3	81.9 ± 40.2	85.3 ± 38.5	
Min–Max	13.0–168.0	32.0–132.0	28.0–159.0	13.0–168.0	
Median (IQR)	87.0 (58.0–113.0)	87.0 (74.0–111.0)	72.0 (62.0–101.0)	87.0 (56.0–118.0)	

* *p* < 0.05.

**Table 2 children-09-00296-t002:** Mother-infant Emotional Connection factors. Descriptive statistics, *n* = 70.

Variable	*n*	Mean	Std Dev	Skewness	Median	Minimum	Maximum
**FACTOR 1. Maternal sensitivity**	70	8.37	3.36	1.39	8	5	20
**FACTOR 2. Emotional concern**	70	17.37	5.31	−0.43	18	5	27
**FACTOR 3. Positive interaction/engagement**	70	6.36	0.87	2.91	6	6	10

**Table 3 children-09-00296-t003:** Mother-infant Emotional Connection factors. Descriptive statistics by maternal milk provision category, *n* = 70 ^1^.

Maternal Milk (MM) Provision Category ^1^	*n*	Variable	*n*	Mean	Std Dev	Skewness	Median	Minimum	Maximum
**1**	**23**	**FACTOR_1**	23	8.78	4.17	1.76	8	5	20
		**FACTOR_2**	23	15.57	5.79	−0.34	17	6	26
		**FACTOR_3**	23	6.87	1.32	1.42	6	6	10
**2**	**9**	**FACTOR_1**	9	8.33	3.5	1.35	8	5	16
		**FACTOR_2**	9	19.44	4.1	−0.17	19	13	25
		**FACTOR_3**	9	6.22	0.44	1.62	6	6	7
**3**	**38**	**FACTOR_1**	38	8.13	2.81	0.47	7	5	13
		**FACTOR_2**	38	17.97	5.06	−0.31	18	5	27
		**FACTOR_3**	38	6.08	0.27	3.25	6	6	7

^1^. Milk provision categories: (1) Exclusive MM, *n* = 23 (reference); (2) Predominant MM, MM on 100% Days Sampled Plus Donor Milk and/or Formula on ≥1 Days, *n* = 9; (3) Mixed, MM on <100% Days Sampled Plus Donor Milk and/or Formula on ≥1 Days, *n* = 38.

**Table 4 children-09-00296-t004:** Spearman correlations (r_s_) between mother-infant emotional connection factors, maternal milk provision category, demographic, and clinical variables.

Variable Label	Study Site (1 = BIDMC, 2 = BWH)	Maternal Age	Primiparous	Days in NICU	Infant Sex	Birth Weight Z Score (Fenton)	Gestational Age	Maternity Employment Leave ^2^	Maternal Education ^3^	Maternal SES	Maternal Race/Ethnicity (1 = Caucasian, 0 = Not Caucasian)
Mother-infant emotional connection factors											
FACTOR 1. Maternal sensitivity (lower scores = greater sensitivity)											
	0.11	−0.02	0.02	−0.17	−0.057	−0.03	0.15	−0.05	0.23	0.21	−0.02
FACTOR 2. Emotional concern (lower scores = greater concern)											
	−0.18	−0.06	−0.15	0.04	−0.11	−0.004	0.04	0.08	−0.2	−0.31 *	−0.09
FACTOR 3. Positive interaction/engagement (lower scores = greater engagement with infant)											
	0.02	0.3 *	−0.15	−0.05	0.06	0.04	0.06	−0.2	0.37 *	0.48 *	0.21 *
Maternal milk provision											
Maternal milk provision category ^1^											
	−0.23	−0.34 *	0.19	−0.03	−0.04	−0.1	0.14	0.15	−0.57 *	−0.57 *	−0.33 *

* *p* < 0.05; ^1^ Milk categories: (1) MM on 100% days sampled, no other milk (donor or formula, OM) fed; (2) MM on 100% days sampled, OM fed ≥1 days sampled; (3) MM fed on <100% days sampled, OM fed ≥1 days sampled; ^2^ 1 = When baby is in the NICU; 2 = When baby comes home; 3 = No maternity leave; 4 = Unemployed/quit job; ^3^ 1= high school diploma/GED; 2 = some college, 3 = college graduate; 4 = advanced degree.

**Table 5 children-09-00296-t005:** Association between milk provision category^1^. and mother-infant emotional connection factors. Unadjusted and adjusted median regression analyses.

	Factor_1 (Maternal Sensitivity)				Factor_2 (Emotional Concern)				Factor_3 (Interaction/Engagement)			
	Low (More)	↔		High (Less)	Low (More)	↔		High (Less)	Low (More)	↔		High (Less)
Milk Provision Category ^1^	β	95% CI		*p*	β	95% CI		*p*	β	95% CI		*p*
Unadjusted												
2	0.00	−3.06	3.06	1.000	2.00	−3.16	7.16	0.442	0.00	−0.27	0.27	1.000
3	−1.00	−3.05	1.05	0.334	1.00	−2.47	4.47	0.567	0.00	−0.18	0.18	1.000
Adjusted by gestational age, maternal education level (categorical), and clustering by site												
2	0.00	−3.38	3.38	1.000	3.00	1.14	4.86	0.002 *	0.00	0.00	0.00	0.017 *
3	2.00	0.76	3.24	0.002 *	3.00	0.00	6.00	0.050	0.00	0.00	0.00	0.866

* *p* < 0.05; ^1^. Milk provision categories: (1) Maternal Breast Milk Only, *n* = 23 (reference); (2) Maternal Breast Milk on 100% Days Sampled Plus Donor Milk and/or Formula on ≥1 Days, *n* = 9; Maternal Breast Milk on < 100% Days Sampled Plus Donor Milk and/or Formula on ≥1 Days, *n* = 38.

## Data Availability

The datasets used and analyzed in this study are available from the corresponding author on reasonable request.

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
