# Peer review of "Maternal Milk Provision in the Neonatal Intensive Care Unit and Mother–Infant Emotional Connection for Preterm Infants"

_children, 2022, doi:10.3390/children9020296_

Round 1

Reviewer 1 Report

This is a well-written clinical article on the relationship between MM provision in the NICU and maternal attachment to her infant in a structured interview validating three factors: 'maternal sensitivity', 'emotional concern' and 'positive interaction/engagement'. The study, well done, is important in highlighting the importance of mothers own milk and its influence on mother-infant bonding.

Furthermore, as the administration of mother's own milk is important for premature infants because of its many -long-term- benefits, early and intensive support for mothers' needs may be helpful in enabling them to express breast milk, especially for black and younger mothers. The improved mother-infant bonding and responsiveness to infants' needs could play an important role in the improved neurocognitive outcome of breast milk feeding. These aspects are sufficiently debated in the discussion.

The following aspects might be added:

  1. Figure 1:

In Figure 1, the flow chart ends with n=300 families/n=362 study offspring. However, only 70 families participated in this study. It would be more understandable to add in a further step why the other families were not included.

Reviewer 2 Report

Thank you for raising an important topic  related to maternal milk and infant mother connection -

Couple of comments - 

1- Line 30 – It should be “< “instead of “ >”

2- Routine kangaroo care was a part of your NICU if the infants were stable enough to be held ? If yes then was the time spent /number of occasion KC was given quantified ?

3-Was here significant number of these patients out-born or all of them were born at these sites .

4- Were Infants with genetic syndromes and extremely sick excluded , such as grade 3-4 IVH .

5- Are investigators planning to publish follow-up developmental data on these infants whether these will result in better neuro developmental outcomes  
